# Hierarchical Exploration of Drying Patterns Formed in Drops Containing Lysozyme, PBS, and Liquid Crystals

**Anusuya Pal** [1,2,*] , **Amalesh Gope** [3,*] **and Germano S. Iannacchione** [1,*]

1 Order-Disorder Phenomena Laboratory, Department of Physics, Worcester Polytechnic Institute, Worcester, MA 01609, USA
2 Department of Physics, University of Warwick, Coventry CV4 7AL, UK
3 Department of Linguistics and Language Technology, Tezpur University, Tezpur, Assam 784028, India
* Correspondence: apal@wpi.edu (A.P.); amalesh@tezu.ac.in (A.G.); gsiannac@wpi.edu (G.S.I.)

**Abstract:** Biological systems, by nature, are highly complex. These systems exhibit diverse hierarchical spatial and temporal features when driven far from equilibrium. The generated features are susceptible to the initial conditions that largely depend on vast parameter space. Extracting information on their properties and behavior thus becomes far too complex. This work seeks to examine the drying kinetics of the drops containing a globular protein (lysozyme (Lys)), phosphate buffer saline (PBS), and thermotropic liquid crystal (LCs). The drying evolution and the morphological crack patterns of these drops are examined using high-resolution microscopy, textural image analysis, and statistical methods. This study observes that the textural parameters can identify the (i) phase separation of the salts present in the PBS and (ii) the LCs' birefringence during the drying evolution. This birefringence activities of the LCs slow down when the initial PBS concentration is increased from 0.25 to $1\times$ despite using a fixed volume of LCs. To comprehend such a surprising effect, the combinations of (i) Lys+PBS and (ii) PBS+LCs are thoroughly examined. A phase diagram is established as a function of initial concentrations of Lys and PBS. The scanning electron microscopic images of Lys+PBS reveal that the tuning between lysozyme and salt concentrations in PBS plays a significant role in determining the morphological patterns. The Lys drops with and without LCs exhibit two distinct regions: the peripheral ring ("coffee-ring") and the central ones. This phase-separated ring formation indicates that the film containing Lys and salts might have formed on top of these LCs in the central region, which reduces the optical response (birefringence) of LCs. A physical mechanism is proposed in this paper to anticipate the redistributions of LCs in a multi-component system such as Lys+PBS+LCs.

**Keywords:** drying; drop; lysozyme; salts; liquid crystals; patterns; texture

## 1. Introduction

Biological systems, by nature, are highly complex. The biosystems' emerging patterns are far more complicated than any other soft system. These features are hard to predict and susceptible, since many biological activities are interconnected and largely depend on vast parameter space. The mesoscopic or macroscopic resulting patterns are primarily sensitive to the initial conditions, including the preparation of the samples, perturbing fields, and so on [1]. A systematic and consistent approach is recommended to explore these systems that require altering one parameter at a time. The complex patterns associated with these bio-systems are due to the local self-assembling interactions between the constituent particles [2]. The transportation of a system from one state (initial bio-colloidal fluid) to another (dried film) through a non-equilibrium (drying) process requires (i) exchanging energy and matter to get acquainted with its micro-environment, and (ii) adjusting the self-assembling interactions between the constituent particles [3].

The research findings of the drying drop community have exhibited numerous studies in the salt-colloidal systems [4–14]. However, the number of findings drops off when the

biological systems in the presence of external salts are concerned. Two globular protein samples, lysozyme and bovine serum albumin (BSA), are predominantly investigated to understand how the patterns are affected when the initial salt concentration is varied [15–22]. Gorr et al. [18] examined lysozyme protein at various concentrations of NaCl. The study by Gorr et al. reported the presence of three distinct regions in the drops with NaCl. The first one is formed in the peripheral ring, where most lysozyme is present. The second one forms different salt structures that occupy the secondary ring area (observed adjacent to the ring), and the final one is observed in the central regions. A similar observation is also reported in BSA-saline protein drying drops by Yakhno [15]. The study by Yakhno concludes that the salt crystals are phase-separated by forming different zones from homogeneous protein film near the periphery to the salt crystals in the central region. Pathak et al. [23] investigated the effects of multiple salts ($MgCl_2$ and KCl) on the BSA patterns. This study reveals that the crystal structures depend on the initial tuning ratio of these salts. Furthermore, a few recent studies examined the formation of these protein-saline drying drops at different elevated substrate temperatures. The findings of these experiments confirmed that the final morphological patterns are mainly dependent on the environmental conditions (higher vs. lower drying rate) [24–26].

New and more sophisticated image processing techniques are developed with the increasing demands of examining the complex images of drying stages. Pattern recognition tools, such as k-means clustering and the k-nearest neighbor algorithm, were applied by Gorr et al. [27] to differentiate the lysozyme-NaCl deposits based on the salts' initial concentration. Carreón et al. [21] applied the first-order statistics (FOS), and the gray level co-occurrence matrix (GLCM) specifying the textural image properties. The study explored the evolution of the final state of drying BSA-lysozyme films in NaCl salts' presence using the FOS and GLCM techniques. Pal et al. [28] also examined BSA drops at different initial concentrations of phosphate buffer saline (PBS). The statistical analysis incorporated in that work showed that these GLCM parameters' horizontal and vertical orientations have a non-significant effect when the pixel displacement is $\leq 1$. The pixel distribution is explored at different PBS initial conditions and regions, such as rim and non-rim regions. The study concluded that the BSA–BSA interactions are dominant over the BSA-saline interactions in the rim regions and vice versa.

On the other hand, many researchers also initiated simplification of these systems by reducing the system's complexity and preparing these protein samples in de-ionized water (without adding any external salts) [17,22,29]. In recent years, Pal et al. provided a new perspective while examining the protein droplets and explored the physics of the drying drops containing optically active particles such as thermotropic liquid crystals [30–32]. The complexity of the multi-component system studied by Pal et al. is more or less similar to the protein-saline drops, since both protein and liquid crystals are non-volatile. The detailed findings along these lines imply that the morphological patterns are altered during the drying process when a fixed volume of liquid crystals is added to different globular proteins (lysozyme, BSA, and myoglobin) [32]. The liquid crystals are distributed randomly in the light-weighted proteins (myoglobin and lysozyme). In contrast, these liquid crystals form umbilical defect structures in heavily weighted proteins such as BSA.

Despite the intense research on protein-saline drying drops, to the best of our knowledge, no systematic study is being performed to understand multiple salts' effects on various concentrated lysozyme protein solutions. It would also be interesting to extend the work from protein+liquid crystals+water to protein+liquid crystals+PBS. Since the liquid crystals (LCs) have polar head groups, and these salts have multiple ionic charges, it would not be surprising to get complex morphological patterns. However, the question here is what is the best possible way to explore the physics of such a complex system? The current paper systematically examines the drying drop consisting of (i) PBS, (ii) LCs+PBS, (iii) lysozyme+PBS, (iv) lysozyme+PBS+LCs, at different initial concentrations. High-resolution (bright-field, cross-field, and scanning electron) microscopy, image-processing tools, and statistical methods are incorporated in our experiments and analysis to explore

their drying evolution and the morphological patterns. In hindsight, this paper attempts to address a few fundamental questions: (i) what are the effects of multiple salts on lysozyme drops with and without LCs? (ii) Do the three regions (as reported in [18]) in lysozyme-saline drops emerge at every initial concentration? (iii) Does it behave uniformly as reported in the BSA drops? (iv) Is it possible to draw fundamental insights on the self-assembly of LCs, proteins, and salts using the proposed image analysis techniques? If so, how important could these insights be?

## 2. Materials and Experimental Methods

The materials used in this study include a lyophilized form of hen-egg white lysozyme, PBS (phosphate buffer saline), and thermotropic liquid crystal (5CB (4-Cyano-4'-pentyl biphenyl)). The lysozyme (Catalog number L6876) was purchased from Sigma-Aldrich, USA. The different concentrations of the PBS were prepared by diluting 1 (purchased from the Fisher BioReagents, USA (Catalog number BP24384)) into 0.75, 0.5, and 0.25×.

The lysozyme has a roughly ellipsoid shape. Its dimension is $3.0 \times 3.0 \times 4.5$ nm$^3$, with an aspect ratio of 1.5. Its molecular mass is $\sim$14.3 kDa [22]. Lysozyme is made up of 129 amino acids. The isoelectric point of lysozyme is 11.1, which allows it to carry a net positive charge. The globular shape and stability of this protein are attributed to the disulfide bridges, hydrogen bonds, and hydrophobic interactions [22]. The $1\times$ PBS solution contains 0.137 M ($\sim$8.0 mg mL$^{-1}$) NaCl, 0.002 M ($\sim$0.2 mg mL$^{-1}$) KCl, and 0.0119 M ($\sim$1.44 mg mL$^{-1}$ of Na$_2$HPO$_4$ and $\sim$0.24 mg mL$^{-1}$ of KH$_2$PO$_4$) phosphates at a pH of $\sim$7.4. The presence of the -cyno groups in 5CB makes it an optically active and polar thermotropic LC. This LC is $\sim$2 nm long and $\sim$0.5 nm in width, with an aspect ratio of 4. It undergoes a phase transition from a crystalline to a nematic phase at 24 °C and from the nematic LC phase to an isotropic phase at $\sim$35 °C [3].

The various amounts, i.e., 100, 75, 50, 35, 25, and 10 mg of lysozyme, are weighed and mixed separately in 1 mL of these PBS solutions. The samples contained the initial concentrations of lysozyme ($\phi_{Lys} = 9.0, 6.9, 4.8, 3.3, 2.4$, and 1.0 wt%) and the initial concentrations of the PBS ($\phi_{PBS} = 1, 0.75, 0.5, 0.25$, and 0×). The $\phi_{PBS} = 0\times$ means the de-ionized water (Millipore, 18.2 MΩ·cm at $\sim$25 °C). Finally, LC (Catalog number 328510, Sigma Aldrich, St. Louis, MO, USA), was heated above its transition temperature, and $\sim$10 µL was added at the fixed lysozyme concentration ($\phi_{Lys} = 9.0$ wt%) as a third component to the different protein-saline drops.

A volume of $\sim$1 µL sample solution is pipetted on a freshly cleaned coverslip (Catalog number 48366-045, VWR, Radnor, PA, USA) under ambient conditions (the room temperature of $\sim$25 °C and the relative humidity of $\sim$50%). The drying evolution is monitored every two seconds only for those samples, where the initial lysozyme concentration is kept fixed ($\phi_{Lys} = 9$ wt%), and the initial PBS concentrations ($\phi_{PBS}$) are varied from 1 to 0× with and without LC droplets. The clock started when the drops were deposited on the substrates. This paper displays the time as $t/t_d$, where the total drying time is denoted by $t_d$, and $t$ is the instantaneous time at which the respective images are captured. The images were captured under $5\times$ magnification using bright-field and cross-polarized optical microscopy (Leitz, Wetzlar, Germany) configured in the transmission mode. An 8-bit digital camera (Model number MU300, Amscope, Irvine, CA, USA) was attached to the microscope to click the top-view images. All these experiments were repeated three times, and all the samples showed the highest reproducibility.

The textural image analysis is performed on the protein drops with and without LCs during the drying process. The oval tool of ImageJ is selected to capture the area of interest in such drops. The first-order statistical image parameters, such as mean gray values (I) and the standard deviation (SD), are extracted using ImageJ [33]. A detailed discussion on the image analysis process is available in [26]. A non-parametric Kruskal–Wallis test (an alternative to the parametric one-way ANOVA test) was preferred to examine possible significant interactions among the different initial PBS concentrations ($\phi_{PBS}$) in terms of these textural parameters (mean and SD). A similar procedure is also adopted

for the samples containing LCs. In the Kruskal–Wallis test, the $\phi_{PBS}$ is the independent factor, with four levels (groups), 0.25, 0.5, 0.75 and 1×, whereas the mean and SD are the dependent variables. The significant level is kept as $p < 0.05$. A pair-wise comparison between all the $\phi_{PBS}$ is drawn using the Bonferroni test in R (Version 3.6.3) embedded with R studio (Version 1.2.1335, RStudio, Inc., Boston, MA, USA). For this, the function *pairwise.wilcox.test()* with the *p.adjust.method = "bonferroni"* is used. The *ggplot2* library and the function *geom_violin()* are used to draw the violin plots.

Scanning electron microscopy (JEOL-7000F, JEOL Inc., Peabody, MA, USA) is used for the samples by varying $\phi_{Lys}$ of $9.0, 4.8, 3.3$, and 1.0 wt% at a fixed $\phi_{PBS}$ of 0.5× and $(\phi_{Lys}, \phi_{PBS}) = (9.0\ \text{wt\%}, 0\times)$. For this, the ~4 nm layer of gold nanoparticles is sputter-coated, and the images are captured at the accelerating voltage of 3 kV and the probe current of 5 mA.

## 3. Results

### 3.1. PBS Drying Drops with and without LC Droplets

The drying evolution of the drops at various initial concentrations of PBS ($\phi_{PBS}$) from 0 to 1× is imaged under bright-field optical microscopy; 0× indicates that the solvent is the de-ionized water. As expected, the drying evolution does not show any residue as the drop dries; however, the deposits' formation changes from 0.25 to 1× (see Figures S1–S3 in the supplementary section). The drying evolution of $\phi_{PBS} = 0.75\times$ is shown in Figure 1a–h. The timestamps ($t$) are calculated as the ratio of the instantaneous time and the total drying time ($t_d$). The water starts evaporating from the drop as soon as it is deposited on the substrate. The height of the drop decreases (see Figure 1a,b how the gray shade changes). The various angles of images and the reduction in drop size confirm that the drop is not pinned to the substrate (see the yellow circular dashed line in Figure 1a–h). The amount of water decreases as the evolution time progresses to such an extent (see Figure 1d) that the concentration of salts increases, and these are carried away with the flow (see Figure 1d–f). Finally, the salts crystallize and fall out of the solution. The final stage demonstrates the evaporation of the trapped water and the formation of a crystal cluster (see Figure 1f–h).

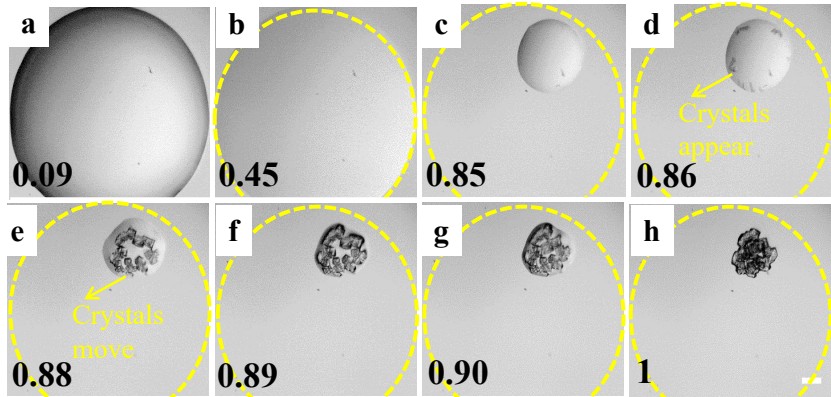

**Figure 1.** Drying evolution of the pattern formation in PBS droplet at the initial concentration ($\phi_{PBS}$) of 0.75× is displayed in (**a–h**). The timestamps are shown with respect to the total drying time ($t_d$) at the bottom left of each image. The yellow dashed circular line shows the drop lining as soon as the drop is deposited on the substrate. The white color in the right corner is a scale bar representing a length of 0.20 mm.

The images of drops containing LC droplets and PBS are captured during the drying process (see Figure 2a–l) using both bright-field and cross polarizing configurations of the optical microscopy. Capturing images consecutively at different configurations ensures the progression of these LC droplets in the PBS is tracked. The process also allows us to compare the drying evolution of the drops with and without LC droplets. The droplets

are mostly carried away towards the periphery of the drop (see Figure 2a–d). A ring-like pattern forms that partially pins the drop on the substrate, unlike the PBS drop (see Figure 1a–h). The drop size does not decrease symmetrically with water evaporation, which can predominantly be seen in Figure 2e. We also observe that a few LC droplets fall out of the solution, i.e., they stay where they are. In contrast, some droplets are carried away with the water (see Figure 2e–h). The salt starts appearing later in the drying process, including the PBS drop (see Figure 2g–i). The LC droplets are observed to be deposited near these crystals as soon as the formation of the salt crystals starts (see Figure 2j–l).

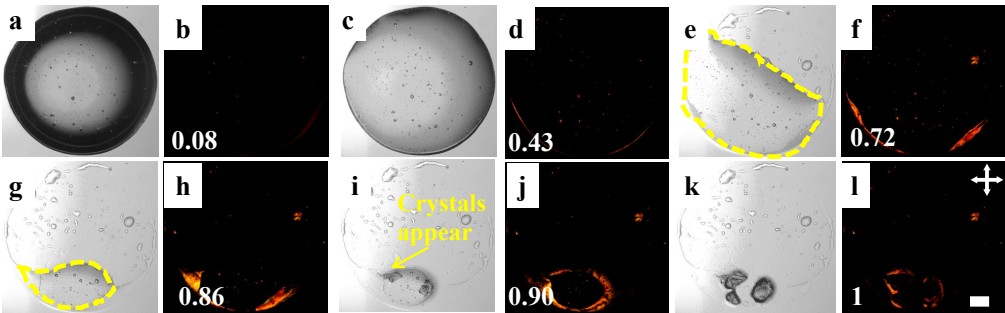

**Figure 2.** Time-lapse images of drying LC+PBS drop at an initial concentration ($\phi_{PBS}$) of 0.75×. The timestamps ($t$) are calculated as the ratio of the instantaneous time and the total drying time ($t_d$), shown at the bottom left of each image. The different stages of the drying process are imaged under bright-field (**a**,**c**,**e**,**g**,**i**,**k**) and crossed polarizing configurations (crossed double arrows). The LC droplets are shown with the bright patches in the crossed configurations (**b**,**d**,**f**,**h**,**j**,**l**). The white color in the right corner is a scale bar representing a length of 0.20 mm.

### 3.2. Lysozyme+PBS Drying Drops

3.2.1. Morphological Patterns at Macro- to Micro-Scales

Figure 3 displays a morphological grid of the samples varying the initial concentrations of Lys ($\phi_{Lys}$) from 9.0 to 1.0 wt% (along the $Y$-axis) and the initial concentrations of the PBS ($\phi_{PBS}$) from 1 to 0.25× (along the $X$-axis). The $\phi_{PBS}$ = 0× embodies the lysozyme solution prepared in the de-ionized water. Though all these deposits show the "coffee-ring" effect [34], diverse patterns are observed for each $\phi_{Lys}$ and $\phi_{PBS}$. The lysozyme films show a mound-like structure when the solution is prepared without external salts. A dimple (or depression) is also noticed within this mound. The mound area gets wider as the $\phi_{Lys}$ increases. The random cracks are only observed in the peripheral ring at ($\phi_{Lys}$, $\phi_{PBS}$) = (1.0 wt%, 0×). However, these cracks spread throughout the film as the $\phi_{Lys}$ increases. The radial and orthoradial cracks promote well-connected (small and large) domains in these drops. Some fringes appear in the concentrated lysozyme samples at $\phi_{PBS}$ = 0×. Many domains in the ring are delaminated, which is predominantly observed at ($\phi_{Lys}$, $\phi_{PBS}$) = (9.0 wt%, 0×). The concentration dependence of these lysozyme drops in the salts' absence is thoroughly explained in [22]. Comparing these patterns reveals that the mound diminishes in the salts' presence. Remarkably, each drop reveals two distinct regions—(i) a peripheral ring ("coffee-ring" [34]) and (ii) a central region. Furthermore, the central regions also disclose two rings at ($\phi_{Lys}$, $\phi_{PBS}$) = (2.4–3.3 wt%, 0.5×) and (2.4 wt%, 0.25×). The presence of multiple rings is also observed in the highly concentrated lysozyme samples, i.e., ($\phi_{Lys}$, $\phi_{PBS}$) = (6.9–9.0 wt%, 0.5×) and (6.9–9.0 wt%, 0.25×). A unique trend is also noticed when $\phi_{Lys}$ is fixed and $\phi_{PBS}$ is varied. For example, at $\phi_{Lys}$ = 1.0 wt%, the ring width decreases with the increasing $\phi_{PBS}$. The central region becomes grainy, the texture becomes darker, and some thread-like structures appear in the central region. The samples at $\phi_{PBS}$ = 1× display a dark texture in the central region and a gray texture in the peripheral ring. Though many drops form the two distinct regions, various textures are observed in the central region. For example, the textures at ($\phi_{Lys}$, $\phi_{PBS}$) = (4.8 wt%, 0.25–0.5×) and (3.3 wt%, 0.25×) are different from ($\phi_{Lys}$, $\phi_{PBS}$) = (2.4–4.8 wt%, 1×). The random small cracks are observed at $\phi_{Lys}$ = 1.0 wt% whereas, the radial cracks predominantly appear in

the peripheral ring as $\phi_{Lys}$ increases in the salts' presence. In contrast, no overall drift is found when these cracks intervene in the central region. For instance, the sample at ($\phi_{Lys}$, $\phi_{PBS}$) = (9.0 wt%, 1×) shows crack patterns, whereas other samples at $\phi_{PBS}$ = 1× do not.

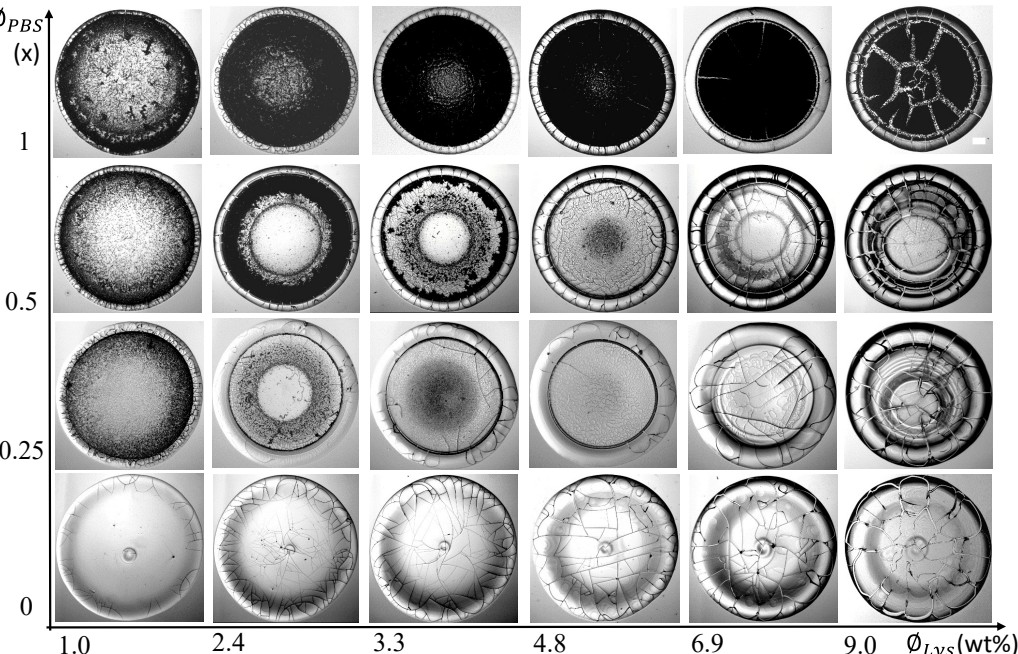

**Figure 3.** Morphological patterns observed in the drying drops of Lys are exhibited here. These patterns are detected for the various initial concentrations of Lys ($\phi_{Lys}$) dissolved in the initial concentrations of the phosphate buffer saline ($\phi_{PBS}$). The $\phi_{PBS} = 0\times$ indicates that the solution is prepared in de-ionized water. The scale bar is of a length of 0.15 mm.

A few lysozyme drops, interestingly, follow different zones while moving from the drop's periphery to the central region. Similar zones are observed in other globular protein drops, viz., BSA [15,35]. A close observation of the samples from peripheral to central regions at ($\phi_{Lys}$, $\phi_{PBS}$) = (9.0 wt%, 0.5×) and ($\phi_{Lys}$, $\phi_{PBS}$) = (6.9 wt%, 0.5×) in Figure 3 display a zone of somewhat homogeneous lysozyme films (or glassy peripheral ring), then a zone of different lysozyme structures, gel-like structures, and the crystalline zone. Therefore, the lysozyme displays different patterns in different volumes of salts. It undergoes phase separation and forms different material properties; for instance, the lysozyme concentration is highest in the peripheral ring and systematically drops towards the central region. On the other hand, the salt concentration is almost null in the periphery but highest as we move towards the central regions. The lysozyme drops in the central region display different cracks in low salt concentrations at $\phi_{Lys} \geq 6.9$ wt%. It is to be noted that these optical images showcase these patterns globally (at macoscale). However, these images do not reveal any microstructural information.

Figure 4I–IV exhibits the microstructures of the various concentrated lysozyme samples at $\phi_{PBS} = 0.5\times$. The sample at ($\phi_{Lys}$, $\phi_{PBS}$) = (9.0 wt%, 0×) is displayed in Figure 4V. The different regions in the central and peripheral regions were emphasized in all these samples. The sample shows a uniform homogeneous texture without external salts (see Figure 4V). In contrast, a distinct texture is observed in the salts' presence (see Figure 4IV). Furthermore, a few non-uniform structures are also uncovered in the crack lines separating the periphery and the central regions. A comparison of the central and the peripheral regions confirms a smooth texture in the peripheral ring. However, some snowflake-like structures appear in the inner ring of the periphery at ($\phi_{Lys}$, $\phi_{PBS}$) = (3.3 wt%, 0.5×) (see Figure 4II). The crystal-like structures are spotted in the zone between the central and peripheral regions at ($\phi_{Lys}$, $\phi_{PBS}$) = (1.0 wt%, 0.5×) (see Figure 4I). These structures; however, do not seem to be too prominent as we move towards the central region of the film. The

central region is mainly replaced with different forms of the dendrite structures; viz, long but thin structures at $(\phi_{Lys}, \phi_{PBS})$ = (3.3 wt%, 0.5×), but then again appears to be shorter but thicker structures at $(\phi_{Lys}, \phi_{PBS})$ = (9.0 wt%, 0.5×) in (see Figures 4II,IV). In these samples, a grainy amorphous layer mostly occupies the middle region (between the peripheral and central regions). On the other hand, it is hard to differentiate this layer between the middle and the central regions at $(\phi_{Lys}, \phi_{PBS})$ = (4.8 wt%, 0.5×) (see Figure 4III).

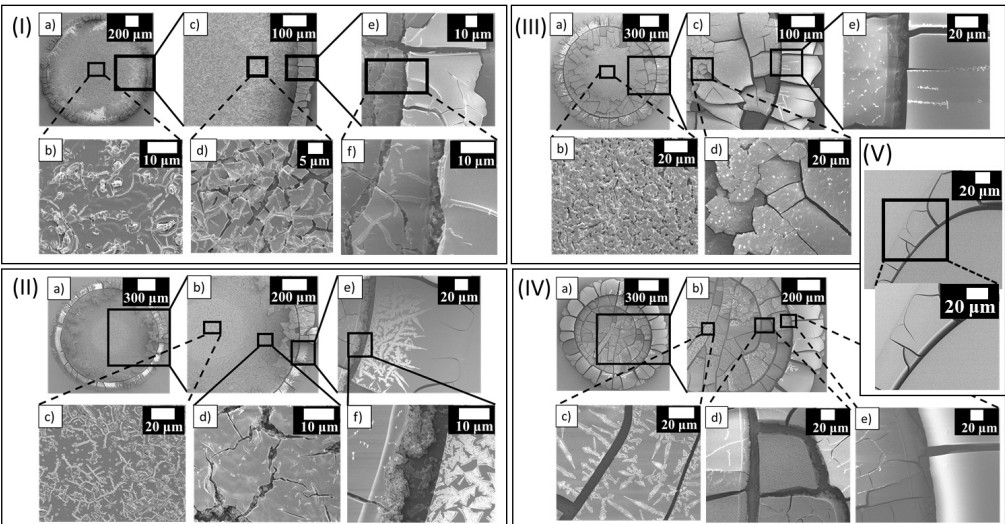

**Figure 4.** Microscopic images of the dried lysozyme samples are displayed in this figure. $(\phi_{Lys}, \phi_{PBS})$ = (1.0 wt%, 0.5×) in (**I**), $(\phi_{Lys}, \phi_{PBS})$ = (3.3 wt%, 0.5×) in (**II**), $(\phi_{Lys}, \phi_{PBS})$ = (4.8 wt%, 0.5×) in (**III**), $(\phi_{Lys}, \phi_{PBS})$ = (9.0 wt%, 0.5×) in (**IV**), and $(\phi_{Lys}, \phi_{PBS})$ = (9.0 wt%, 0×) in (**V**). Different length scales are shown as the scale bars in the upper-right corner of each image.

### 3.2.2. Qualitative and Quantitative Analysis of Drying Evolution

To understand how these distinct structures appear in different regions, we examine the drying evolution and dried morphology by keeping $\phi_{Lys}$ at 9.0 wt%, and only the $\phi_{PBS}$ is varied from 0.25 to 1×. Figure 5A–D describes the drying evolution of the lysozyme drops prepared with different PBS concentrations. The timestamps are calculated as the instantaneous time divided by the total drying time ($t_d$). A uniform gray texture with a dark peripheral band is observed in all the drops when the first image is captured of the drying process (see Figure 5A). The fluid front moves from the periphery to the central region with time progression (see Figure 5B). Surprisingly, the texture of the front movement changes in the salts' presence. Once the peripheral ring emerges, the grainy texture develops in the central region. A clear distinction is visible at the interface of the inner peripheral ring. At $\phi_{PBS} = 0.25\times$, the formation of the dark texture is not predominantly observed; however, the darkness increases as the $\phi_{PBS}$ rises. Simultaneously, the cracks propagate from the periphery toward the center. However, the propagation is not smooth, unlike $\phi_{PBS} = 0\times$, and the propagation is interrupted in the presence of the salts. The mound-like structure begins in the last stage of this fluid front movement, but the salts' presence diminishes its formation in the central region. The multiple rings are found as the front's radius gets smaller (see Figure 5B,C). After the visible drying process, the final morphological patterns are captured in Figure 5D.

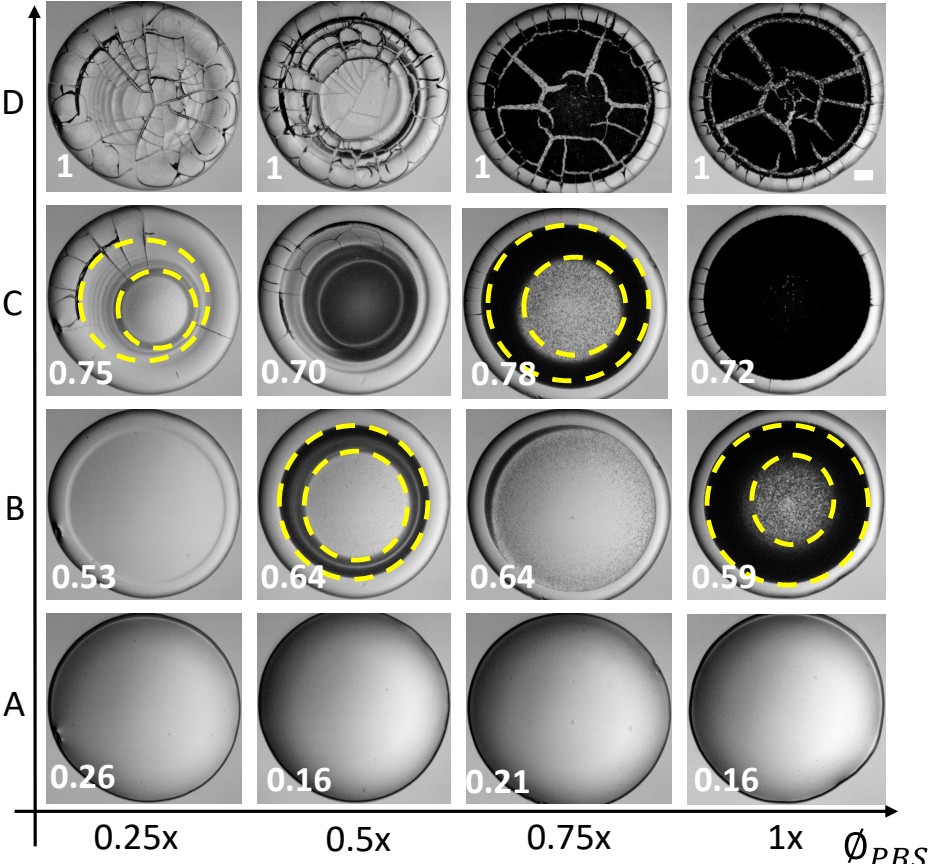

**Figure 5.** The time evolution of lysozyme drops ($\phi_{Lys} = 9$ wt%) at various initial concentrations of salts in PBS ($\phi_{PBS}$) during the drying process is displayed in (**A**–**D**). The instantaneous time is divided by the total drying time ($t_d$) to calculate the timestamps, as shown in the bottom left of each image. The white rectangle represents a scale bar of 0.15 mm length in the top right.

Figure 6I–IV shows the quantitative analysis of the textural evolution during the drying process. The first-order statistical parameters, the mean gray values ($I$), and the standard deviation ($SD$) are exhibited as a function of the drying time (in seconds) at $\phi_{PBS}$ ranging from 0.25 to $1\times$. It is to be noted that these parameters describe the gray level distribution of the image's pixel intensity. The $I$ defines the averaged values, whereas the $SD$ illustrates the textural complexity. The $I$ stays nearly constant at the beginning of the drying process. It reduces and starts fluctuating (marked with a star in Figure 6I–IV). In contrast, the $SD$ decreases linearly until $t/t_d \sim 0.6$ and rapidly rises for $t/t_d \sim 0.6$–0.7. It first decreases, and grows again (marked with a star in Figure 6I–IV). Finally, both the $I$ and $SD$ saturate in the last phase of the drying evolution. Interestingly, both the $I$ and $SD$ exhibit significant changes when $t/t_d \sim 0.6$–0.9. The $I$ decreases, and the $SD$ increases when we prepare the drops by adding more PBS. For instance, the $I$ reduces from $\sim$50 to $\sim$20 a.u. at $\phi_{PBS} = 1\times$, whereas it only varies $\sim$15 a.u. when $\phi_{PBS} = 0.25\times$. The images (see Figure 6I–IV) show that considerable variation in these textural parameters ($I$ and $SD$) occurs when the dark textured fluid front moves in the central region.

Figure 7I,II shows the textural parameters, mean, and standard deviation (SD) at different initial concentrations of PBS ($\phi_{PBS}$). The samples considered for the statistical analysis include the protein drops at a fixed initial concentration ($\phi_{Lys} = 9$ wt%) that do not contain any liquid crystals (LCs). The $\phi_{PBS}$ is varied from 0.25 to $1\times$. The violin plots in Figure 7I,II combine the box plots with the density traces (viz., the peaks, valleys, lengths, etc.). This density trace is vertically plotted to both sides of the box plots. These plots display the statistical description in which a rectangle represents the second and third quartiles, and a line inside that box indicates the median value. The traces are symmetrically

drawn; only the directions are reversed to observe the density magnitude in a better way. The box plots and the density traces allow us to get a quick and insightful comparison of the data distributions. However, these plots do not provide any information about their sample size [36]. Here, the median at all $\phi_{PBS}$ has been sifted for the textural parameters, mean, and the SD. The mean shows a bi-modal distribution for $\phi_{PBS}$ = 0.25 and 0.5×, and somewhat normal distribution is observed in the range of 0 to 20 (a.u.). However, a positive skewness is observed in the range of 40 to 60 (a.u.) in the $\phi_{PBS}$ = 0.75 and 1× (see Figure 7I). The SD reveals a uni-modal distribution at $\phi_{PBS}$ = 0.75×. The SD data, nonetheless, are skewed in the range of 10 to 15 a.u. (see Figure 7II). Both the textural parameters confirm a non-normal distribution. The skewness also confirms that the mean and the SD are not equally distributed throughout the drying process (see Figure 6I–IV). Therefore, a non-parametric Kruskal–Wallis test is performed. A detailed report of the pair-wise comparison is shown in Tables S1 and S2 of the supplementary section. It must be noted that the visual display of Figure 6I–IV also indicates possible differences in the textural parameters at each $\phi_{PBS}$. The proposed statistical test facilitates quantitatively distinguishing and confirming that the mean and SD are different at each $\phi_{PBS}$.

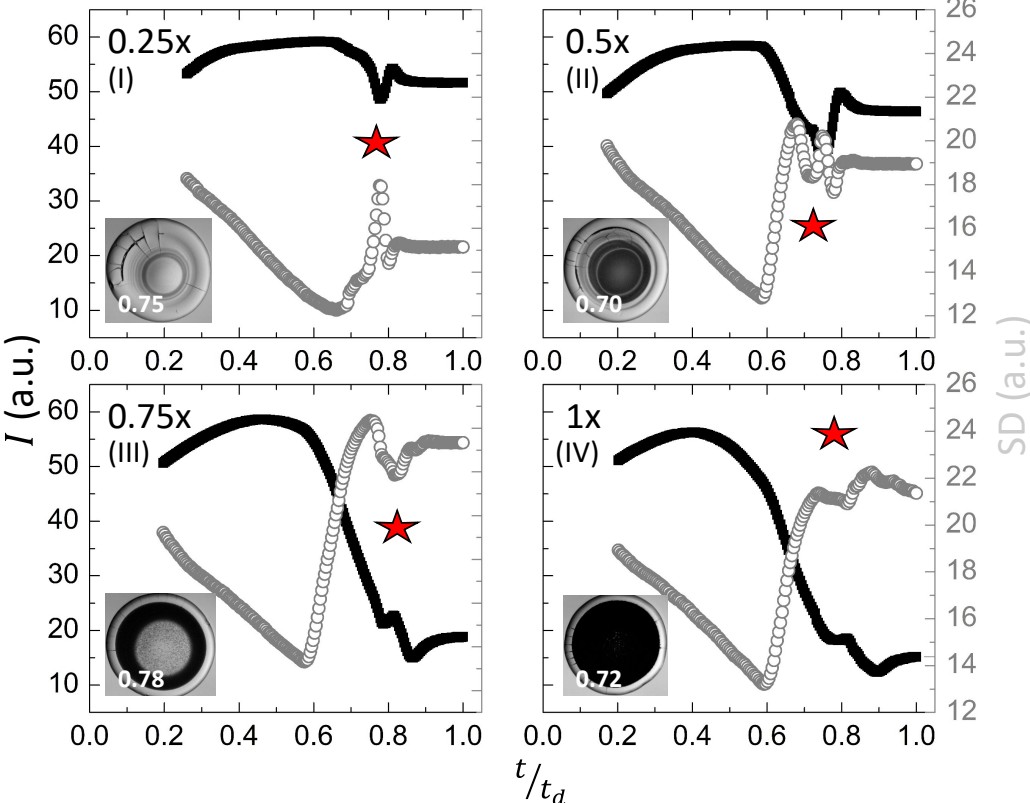

**Figure 6.** Textural analysis of the lysozyme drops ($\phi_{Lys}$ = 9 wt%) at various initial concentrations of salts in PBS ($\phi_{PBS}$) during the drying process is displayed in (**I–IV**). The *x*-axis displays $t/t_d$, where the total drying time is denoted by $t_d$, and $t$ is the instantaneous time captured by the respective image. The left-*Y* axis (shown in back color squares) and the right-*Y* axis (shown in gray color circles) in each graph describe the mean gray values (*I*) and the standard deviation (*SD*) in arbitrary units (a.u.), respectively. The star symbol indicates the fluctuations in these textural parameters.

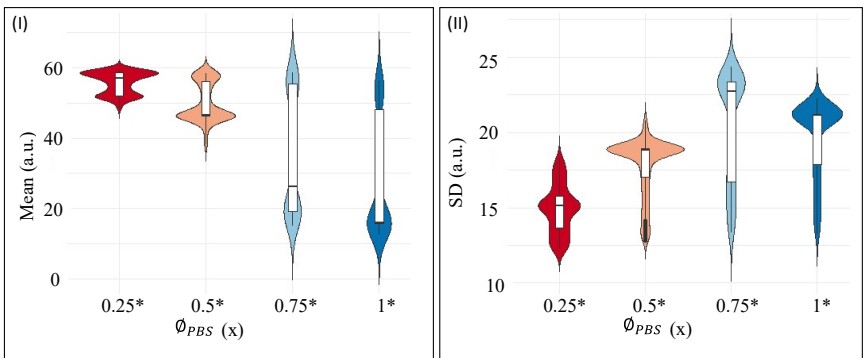

**Figure 7.** Violin plots for the first-order textural parameters—(**I**) represents the distribution of Mean, and (**II**) displays the distribution of Standard Deviation (SD) of the protein drops at different initial concentrations of PBS ($\phi_{PBS}$). The significant $\phi_{PBS}$ is marked with an asterisk (*). The protein drops have a fixed initial concentration of 9 wt% that do not contain any liquid crystals (LCs).

### 3.3. Lysozyme+PBS Drying Drops with LC Droplets

The drying evolution of the lysozyme+PBS drops with LC droplets at different $\phi_{PBS}$ of 0 to $1\times$ under crossed polarizing configurations is shown in Figure 8A–D. The $\phi_{Lys} = 9.0$ wt%, and the volume of LC (i.e., ~10 μL) are kept constant, and only the $\phi_{PBS}$ is varied from 0 to $1\times$. The timestamps ($t$) are calculated by dividing the instantaneous time by the total drying time ($t_d$). Here, we are interested to know how the optical activities or the birefringence properties of these LC droplets are influenced due to the presence of the salts. The $\phi_{PBS} = 0\times$ denotes no external salt, whereas the initial salt concentration systematically increases from 0.25 to $1\times$.

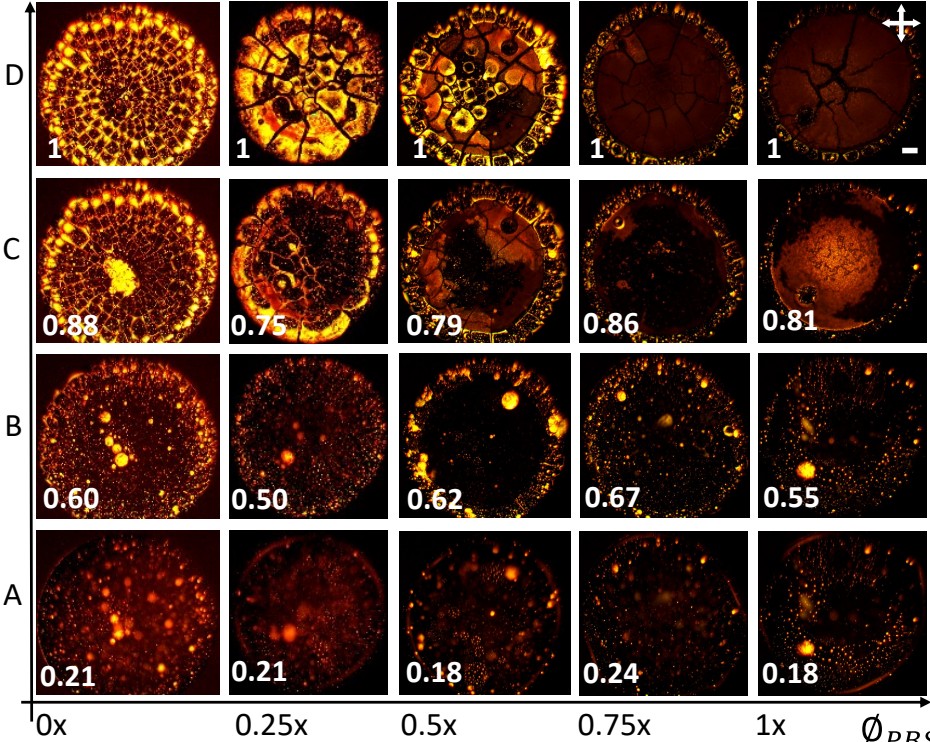

**Figure 8.** Time-lapse images of drying lysozyme+PBS drop with LC droplets at the initial concentration ($\phi_{PBS}$) of 0 to $1\times$ are displayed in (**A–D**). The timestamps are shown at the bottom left of each image. The images are captured under crossed polarizing configuration (crossed double arrows). The white color in the right corner is a scale bar representing a length of 0.20 mm.

The LC droplets are carried away towards the periphery of the drop once this is deposited on the substrate, irrespective of the presence of the salts (see Figure 8A,B, $\phi_{PBS} = 0\times$). It is to be noted that the volume of the LC added in the drop is of equal quantity. Therefore, the image captured after depositing the drop on the substrate is expected to be similar. However, a slight variation is found as we increase the $\phi_{PBS}$. In the absence of the salts, these droplets follow the crack lines (see Figure 8C, $\phi_{PBS} = 0\times$). Subsequently, each cracked domain is partially filled with these droplets. Each domain has a central dark region surrounded by a bright region (see Figure 8D, $\phi_{PBS} = 0\times$). A detailed description of the drying evolution of lysozyme drop with LC droplets at $\phi_{PBS} = 0\times$ is available in [30]. On the other hand, the crack lines and the filling up of the domains are faintly observed at $\phi_{PBS} = 0.25\times$. Interestingly, this process diminishes as we increase $\phi_{PBS}$, and a new feature evolves. The central part of the drop changes from the dark to the bright grainy region (see Figure 8C). The bright regions get brighter at 0.5 and $0.75\times$, whereas these turn darker at $\phi_{PBS} = 1\times$ (see Figure 8D). Finally, two distinct regions, i.e., peripheral and central regions, can be identified in the presence of salts. The number of cracks decreases, increasing the size of the cracked domains as we keep on adding more salts into the system.

The visual observation is quantified using the textural image analysis of the first-order statistics (mean gray values (I in a.u.) and standard deviation (SD in a.u.)) (Figure 9I–IV at different $\phi_{PBS}$). The I and SD remain constant in the initial drying stage irrespective of the salt concentrations. However, their transient behavior at $t/t_d \sim 0.7$–0.85 is interesting. At $\phi_{PBS} = 0.25\times$, both SD and I rapidly increase and saturate. The I increases, whereas SD shows fluctuating behavior at $\phi_{PBS}$ of 0.5 to $0.75\times$. This increase in I, indeed, gets smaller as we increase $\phi_{PBS}$. For instance, it grows from $\sim$15 to $\sim$75 a.u. at $\phi_{PBS} = 0.25\times$ but only increases to $\sim$25 a.u. at $\phi_{PBS} = 0.75\times$. The $\phi_{PBS} = 1\times$ shows different behavior from the rest. The I decreases again at $t/t_d \sim 0.8$ and eventually saturates. Overall, we can say that the birefringence (or the optical activity) of the LC droplets reduces as we increase the initial concentration of PBS.

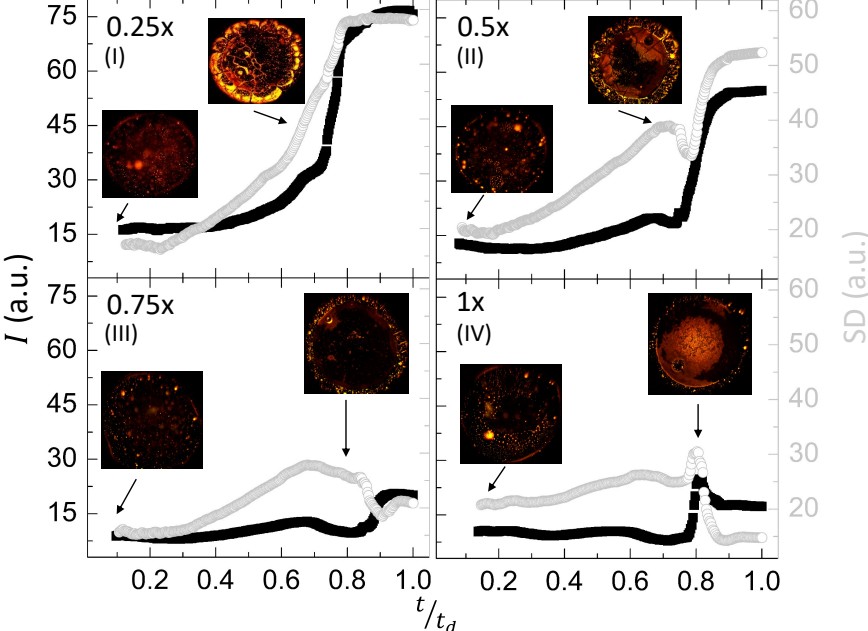

**Figure 9.** Textural analysis of the lysozyme drops ($\phi_{Lys} = 9$ wt%) with LC droplets at various initial concentrations of salts in PBS ($\phi_{PBS}$) during the drying process is displayed in (**I–IV**). The x-axis displays $t/t_d$, where the total drying time is denoted by $t_d$, and $t$ is the instantaneous time captured by the respective image. The left-Y (shown in back color squares) and the right-Y axes (shown in gray color circles) in each graph describe the mean gray values ($I$) and the standard deviation ($SD$) in arbitrary units (a.u.), respectively.

Comparing Figures 6 and 9I–IV, it is found that the textural analysis identifies different significant activities in the drop. Interestingly, this is captured for $t/t_d \sim 0.6$–0.85 during the drying process in the lysozyme drops with and without LC droplets.

Figure 10I,II displays the violin plots at different initial concentrations of PBS ($\phi_{PBS}$) for the textural parameters, mean and standard deviation (SD). The samples considered for the statistical analysis include the protein drops at a fixed initial concentration ($\phi_{Lys} = 9$ wt%), and contain liquid crystals (LCs). The $\phi_{PBS}$ is varied from 0.25 to 1×. Similar to Figure 7I,II, the median at all $\phi_{PBS}$ has sifted for both the textural parameters, mean and SD; however, it is not possible to make one-to-one comparisons between the protein drops with and without LCs (viz., between Figures 7 and 10), since the images without LCs are captured under bright-field configuration. In contrast, the images with LCs are captured under crossed polarizing configurations. However, both the textural parameters confirm a non-normal distribution. The skewness suggests that the mean and SD are not equally distributed throughout the drying process [also evident from Figure 9I–IV]. A detailed report of the pair-wise comparison is shown in Tables S3 and S4 of the supplementary section.

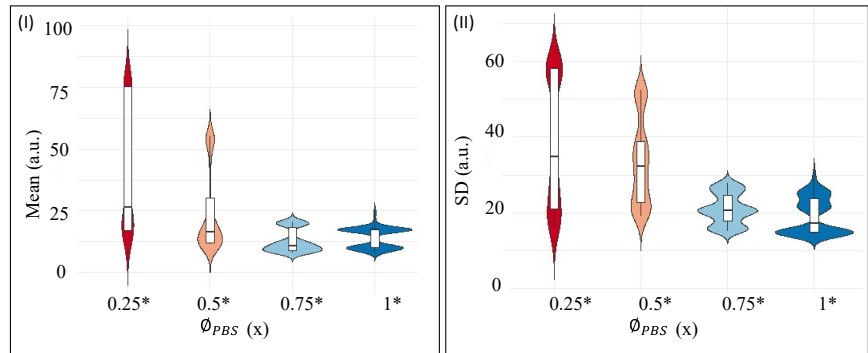

**Figure 10.** Violin plots for the first-order textural parameters. (**I**) represents the distribution of Mean, and (**II**) displays the distribution of Standard Deviation (SD) of the protein drops at different initial concentrations of PBS ($\phi_{PBS}$). The significant $\phi_{PBS}$ is marked with an asterisk [*]. The protein drops have a fixed initial concentration of 9 wt% and contain a fixed volume of liquid crystals (LCs).

## 4. Discussions

The height and the contact angle start reducing as soon as the drops are pipetted on the substrate. The non-uniform textural gradient in the optical images (for example, Figure 5A,B) indicates that the height of the drop initiates the decreasing process in the initial drying stage itself. These drops are of a spherical-cap shape and are partially wet (checked with a goniometer). The curvature of these drops induces the highest mass loss near the periphery compared to their central region. The drops get pinned to the substrate, and the lysozyme particles are transported through the outward capillary radial flow to compensate for this loss. The process leads to the popular "coffee-ring effect" [34] that is also observed in other bio-colloids as well [20,22]. Furthermore, most LC droplets are carried away towards the periphery of the lysozyme drops.

With the progression of the evolution time, the fluid front recedes from the periphery to the central region in the lysozyme drops. Concurrently, the contact angle reduces, contrary to the results reported in [19]. The deposits in the crack lines (see Figure 4) indicate a discontinuity in the crack lines at the ring interface (also evident in Figure 5B,C). It is to be noted that the evaporation of a significant amount of water initiates the formation of salt crystals at this time. Since the images were taken in the transmission mode, the thick film gives a dark texture. The dark textured front starts engulfing the central region (a similar phase transition phenomenon reported in [15]). A significant fluctuation in the textural evolution (see Figure 6I–IV) is also observed at this phase. The complexity (SD) increases as the salt crystals (inhomogeneities) appear.

The crack propagation of the pinned drops relieves the mechanical stress. The appearance of the salt crystals in different lysozyme concentrations affects the crack formation

process (see Figure 3). It also alters the interaction between the lysozyme particles and changes their aggregation and precipitation processes (samples with and without adding external salts, evident from Figures 3 and 4). The images captured at high and low initial concentrations of PBS ($\phi_{PBS}$) at (i) fixed and (ii) different initial concentrations of lysozyme ($\phi_{Lys}$) reveal contrasting morphological characteristics. For example, the dried drops formed at $\phi_{Lys}$ = 1 wt%, ($\phi_{PBS}$) = 0.25×, exhibit a rough texture with block salt crystals in the central region. On the other hand, a very thin "coffee ring" and small needle-shaped structures appear in the peripheral region (see Figures 3 and 4(I,e)). In contrast, deposits with higher $\phi_{PBS}$ contain a thick "coffee ring" near the edge, and large rosette-like crystal structures appear in the central region (see Figure 4(IV,c)). These contrasting morphologies emerge from the different nucleation and super-saturation points during the drying process. Interestingly, this appearance of different structural morphology is not limited to the protein drops but is also reported to be observed in many polymer-saline drops [14].

This study also demonstrates that the three prominent regions (as reported in [18]) may or may not surf in the lysozyme drops when the salts are in the solution. This study, therefore, argues that the occurrence of three distinct regions should not be generalized for lysozyme+PBS drying drops. We further argue that the three distinct regions in the lysozyme drops depend on the relative initial concentrations of both lysozyme and salts. The only variation of the salt content might not provide us with a clear picture. Figures 3 and 4 show that the chemistry between multiple salts and lysozyme at various initial concentrations (both lysozyme and salts) is the crucial factor in determining morphological patterns. In this context, we established a phase diagram based on the initial concentrations of lysozyme ($\phi_{Lys}$) and PBS ($\phi_{PBS}$). Figure 11 exhibits the various phases that are colored uniquely. For example, the orange color describes a phase where the drops always show one ring that divides the central region from the periphery. Two representative images are shown in this phase to illustrate that the central region's grainy texture changes to black as we increase the $\phi_{PBS}$. However, we are ignoring this gradual textural change and counting all in one phase. Though the phase of white color also shows one ring, we used a different color to indicate that the texture of the central regions of both the phases is not the same. The phase of blue color suggests that the drops have multiple rings in the central region in addition to the peripheral ring (illustrated with the dashed red colored line in Figure 11). The green color indicates the presence of only two rings (one in the central region and another in the peripheral ring). The hatched lines for $\phi_{PBS} = 0\times$ indicate that the drops without adding any external salts are totally different from those with PBS. The presence of a mound-like structure (illustrated with the solid red colored line in Figure 11) in the central region confirms that the underlying physics of such drying-mediated patterns are different for the drops with and without salts. Different phases in Figure 11 also confirms that the hierarchical structures that are formed by the aggregation of the lysozyme and salts are not directly correlated with their initial concentrations. If so, we would have the same phase as we double their concentrations. For instance, that phase at ($\phi_{Lys}$, $\phi_{PBS}$) = (1.0 wt%, 0.25×) is not the same as the ($\phi_{Lys}$, $\phi_{PBS}$) = (2.4 wt%, 0.5×) or ($\phi_{Lys}$, $\phi_{PBS}$) = (2.4 wt%, 0.5×) is not the same phase as ($\phi_{Lys}$, $\phi_{PBS}$) = (4.8 wt%, 1×).

However, it is hard to predict how the adsorption process of various particles (lysozyme and salts) occurs during the drying process. The SEM images of lysozyme+PBS drops (see Figure 4I–IV) indicate that the lysozyme forms a film on the substrate during the initial drying stage. The salts are phase-separated during the mid-drying stage, forming two different regions. The peripheral region contains most lysozyme particles, whereas the central region contains lysozyme and salts. It is worthy of mentioning here that salt crystallization is a natural process formed due to the evaporation of a large volume of water from the system (similar to the salt formation in the sea beds). It can be assumed that the lysozyme particles in the film act as the nucleation points and initiate distinct types of salt crystallization. For instance, some salts crystallize to form dendrite-like structures on the protein films, whereas some appear as snowflakes, sword-like structures, etc. Some aggregated globules are also present in the crack lines (see Figure 12(AI,AII)). In this context, a

plausible mechanism can be drawn in terms of the protein charges, affinity of salts, surface properties, etc. It can be assumed that the globule nature of these proteins is maintained. As a result, the hydrophobic residues of the proteins are buried inside the protein core, and both positively and negatively charged residues on the protein surface are exposed. It is to be noted that the coverslip (substrate) is negatively charged, and the overall charge of lysozyme is positive. Thus, the positively charged residues adsorb the substrate [22]. The probability of altering the overall interaction of lysozyme with the substrate is very low.

On the other hand, there is a high chance of different lysozyme–lysozyme and/or lysozyme–salts interactions. This is because the interactions between opposing surface charges and hydrophobic regions binding these globular proteins might be favored. Similarly, the interaction between the lysozyme particles and the solvent produces a charged layer (Stern layer of counter-ions) [37]. This layer comprises water molecules and/or different dissociating ions present in PBS (viz., $Na^+$, $K^+$, $Cl^-$). The oxygen and hydrogen atoms present in the water are likely to be attached to the protein surfaces' positively and negatively charged ions, respectively, due to their electrostatic interactions. The further progression of the drying process leads to the modification of the proteins' hydration shell. Finally, the evaporation of a significant amount of water from the drops facilitates the emergence of the super-saturation or nucleation points.

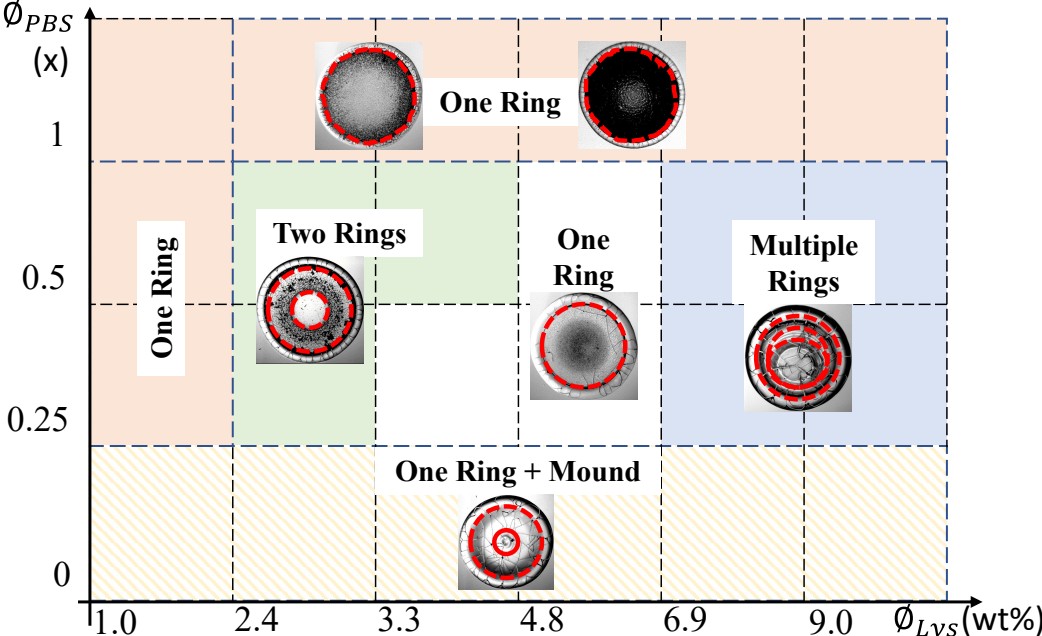

**Figure 11.** A phase diagram is established for the drying-mediated patterns observed in Figure 3. Different phases, indicated by distinct color shades—for example, one, two, and multiple rings in the drops—are presented as a function of various initial concentrations of Lys ($\phi_{Lys}$ along the X-axis) and the phosphate buffer saline ($\phi_{PBS}$ along the Y-axis). The hatched lines for $\phi_{Lys} = 0\times$ indicate that the drops without adding any external salts are different from the drops with PBS by the presence of the mound-like structure in the central region. The representative image in each phase is provided where the dashed red-colored lines mark the rings, and the solid line depicts the mound-like structure.

The addition of LC droplets in the lysozyme drops makes the scenario far more complicated. Let us look at the drops prepared in the de-ionized water ($\phi_{PBS} = 0\times$). The LCs fill the crack lines of the drop at random. Once the protein-cracked domains buckled up due to mechanical stress (see Figure 3, ($\phi_{Lys}$, $\phi_{PBS}$) = (9 wt%, 0×)), the randomly distributed LCs are sucked underneath these domains (see Figure 12(BIII,BV)). Accordingly, we can confirm that the bright regions observed under crossed polarized configurations are the randomly oriented LCs distributed underneath each lysozyme domain. The dark region in each domain corresponds to the attached protein layer that is not optically active.

A detailed discussion on this is available in [30]. It is to be noted that the aspect ratios of lysozyme and LC are 1.5 and 4, respectively. Therefore, it can be assumed that there is a probability of a size effect on the crack formation as reported in other multi-component systems [22,38]. The bright-field images of the lysozyme drops with and without LCs show that the cracks are more ordered in the presence of LCs, whereas a chaotic system is observed in their absence [31]. However, extracting the exact physical mechanism due to the difference in their sizes is not within the scope of this paper.

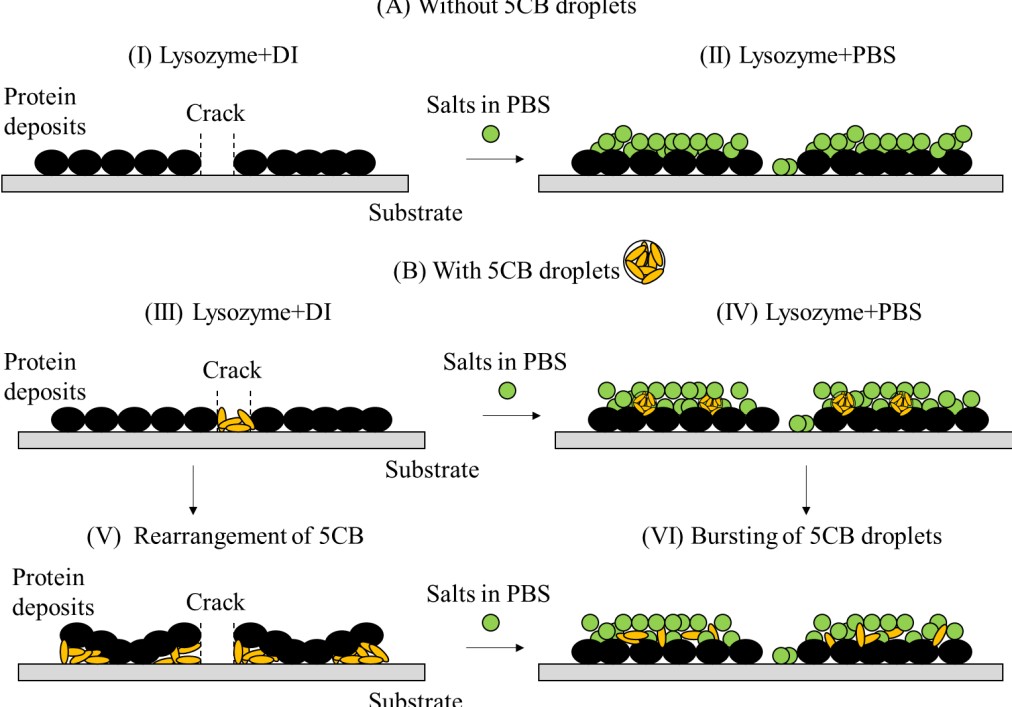

**Figure 12.** A schematic representation of lysozyme drops prepared in different concentrations of PBS ($\phi_{PBS}$) with and without LC droplets is shown in this figure. (**A**)(I,II) depicts the changes in the central region of the drop when we add concentrated $\phi_{PBS}$. (**B**)(III–VI) illustrates how the LC droplets affect the distribution of proteins, salts, and the final morphological patterns.

A proper explanation of the drying mechanism in the presence of salts, lysozyme, and LC droplets can be drawn from Figures 5–9. The LC droplets behave more or less like 0× at the lower salt concentration, $\phi_{PBS} = 0.25\times$ (see Figures 8 and 9I). The LC comprises organic molecules with (i) a rigid core of two phenyl groups and (ii) a side chain of a cyano group (-CN⁻), making it polar. The cyano group in the LCs interacts with the dissociating positively charged ions of the PBS (Na⁺, K⁺, etc.) and the positively charged residues of the lysozyme. This interaction possibly impacts the packing of both lysozyme particles and LC droplets as more water evaporates during the drying process. Interestingly, we can see that the crack domains are larger than that of 0×, which is true for all the protein drops in the presence and absence of LCs. This indicates that the interactions between the salts–salts, salts–proteins, and proteins–proteins are rigid, and that they want to be together. It might also be the case that the presence of the salts has increased the film height, so the mechanical stress (via crack formation) acts differently from the drops without salts. This observation did not change when we added LCs into the system. This means LCs might not increase the film height but only affect the packing of the particles in the system. In other words, the LC droplets are somehow trapped in the layer between lysozyme particles and salts in the central regions. The evaporation of a further volume of water at the later drying stage ($t/t_d \sim 0.7$–0.8) propagates the bursting and random distribution of these droplets (see Figure 12(BIV,BVI)). Unlike 0×, there is an additional salt layer on top of the

LC distribution. In contrast, the peripheral region of the lysozyme+PBS+LC drop mostly contains the lysozyme particles, which drive the LCs to fill these cracked domains. Since the LCs do not have enough volume to fulfill the cracked domains, and the water leaves the system by that time, the LCs stay where they are. The whole process thus provides a plausible explanation of the reduced birefringence intensity (I under crossed polarizing configuration) when the $\phi_{PBS}$ was increased despite having a fixed volume of LCs (see Figure 9I–IV).

## 5. Conclusions

This paper reveals that the tuning between lysozyme and salts is essential in determining the morphological patterns. It rasters the parameter space by varying the initial concentrations of both lysozyme and PBS salts. The findings of this paper thoroughly explain the hierarchical complexity of the multi-component system consisting of (i) PBS, (ii) PBS+LCs, (iii) lysozyme+PBS, and (iv) lysozyme+PBS+LCs at different initial concentrations. The textual evolution indicates that the interactions between different lysozyme particles during the drying process are dependent on the amount of the salts present in PBS. It also shows that the occurrence of three distinct regions is not the general characteristic of lysozyme-saline droplets; rather, it depends on the relative initial concentrations of lysozyme and salts. Different phase-separated zones reported in the BSA drops are also observed in some lysozyme samples. This implies that the phase separation of salts, and lysozyme particles in different structural forms, might not be specific to the individual protein characteristics; instead, this phase separation is associated with all the globular proteins. Different pattern-recognized phases established by the phase diagram also confirm that the hierarchical structures formed by the aggregation of the lysozyme and salts are not directly correlated with their initial concentrations. This study proposes a plausible mechanism supporting the optical behavior (birefringence) of the LCs that relates the morphological patterns at the macroscale to the different interactions at the microscale. The experimental findings of this study anticipate the redistribution of these LCs. This study, however, does not advocate a particular experimental protocol or technique that can exactly explore the complex interactions at the interface, viz., protein–protein interactions, or the distribution of LCs, etc. Nonetheless, this study motivates researchers to develop in situ experimental set-ups to examine and explore the clear-cut complex simultaneous events during such non-equilibrium processes in the near future.

**Supplementary Materials:** The following supporting information can be downloaded at: https://www.mdpi.com/article/10.3390/pr10050955/s1, Figure S1. Drying evolution of PBS drop at the initial concentration ($\phi_{PBS}$) of (I) 0x and (II) 0.25x. Figure S2. Drying evolution of PBS drop at the initial concentration ($\phi_{PBS}$) of (I) 0.5x, and (II) 0.75x. The scale bar of length 0:15 mm. Figure S3. Drying evolution of PBS drop at the initial concentration ($\phi_{PBS}$) of 1x. Table S1. Pairwise comparison of mean among all PBS concentrations ($\phi_{PBS}$) for the drops at a fixed lysozyme initial concentration. These drops do not contain any liquid crystals (LCs). Table S2. Pairwise comparison of SD among all PBS concentrations ($\phi_{PBS}$) for the drops at a fixed lysozyme initial concentration. These drops do not contain any liquid crystals (LCs). Table S3. Pairwise comparison of mean among all PBS concentrations ($\phi_{PBS}$) for the drops at a fixed lysozyme initial concentration. These drops contain liquid crystals (LCs). Table S4. Pairwise comparison of SD among all PBS concentrations ($\phi_{PBS}$) for the drops at a fixed lysozyme initial concentration. These drops contain liquid crystals (LCs).

**Author Contributions:** All the authors were actively involved in preparing the final manuscript. The experiments were designed and performed by A.P. The interpretation of the results was carried out by A.P and A.G. The images were drawn, quantified, and computed by A.G. and A.P. G.S.I. supervised the work. A.G. also edited the final version of the manuscript. All authors have read and agreed to the published version of the manuscript.

**Funding:** This work is supported by the Department of Physics at WPI, USA, and the University of Warwick, UK. This study was partially funded by Leverhulme Trust (Grant No. RPG-2018-345).

**Institutional Review Board Statement:** Not applicable.

**Informed Consent Statement:** Not applicable.

**Data Availability Statement:** Not applicable.

**Conflicts of Interest:** The authors declare no conflicts of interest.

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
