# Peer review of "Hierarchical Exploration of Drying Patterns Formed in Drops Containing Lysozyme, PBS, and Liquid Crystals"

_processes, doi:10.3390/pr10050955_

Round 1

Reviewer 1 Report

I reviewed the manuscript and I believe that this research was done very well. As for the research itself, its methodology applied procedures, and obtained results, I have no objections. However, the way in which all this is presented in the manuscript is very difficult to follow and makes the whole story confusing, although there is no reason for that. My only remark concerns the way of presentation, which is very concise and therefore difficult to follow. Maybe the text should be broken down into short explanations, maybe some values should be presented in tabular form, etc., I don't know, but to fully understand the whole material, I needed to read the paper many, many times. You may or may not accept the remark, it certainly does not affect my opinion that the work is good and should be accepted for publication.

Author Response

see attached the file

Reviewer 2 Report

This paper lacks many things. Just look at how a drop evaluate with respect to concentrations. If Authors want to do drying they should be related to the time domain. Also as given in the introduction only microscopy can be seen. Where are the image processing and statistical analysis. You need to incorporate those to the paper to be accepted. Please rectify those lapses and resubmit.

Author Response

see attached the file

Round 2

Reviewer 2 Report

General comments;

Please go through the manuscript again and do minor corrections in typos and english grammar.

Technical comments:

Please improve the clarification of methods.

Author Response

Thanks for the comments. We revised our manuscript thoroughly a few times. We have also added a figure on the phase diagram for clarity and to get more fundamental insights. Please see the highlighted texts in the attached file. 
